# Is Herbertsmithite far from an ideal antiferromagnet? Ab-initio answer including in-plane Dzyaloshinskii-Moriya interactions and coupling with extra-plane impurities

Flaurent Heully-Alary[1], Nadia Ben Amor[1], Nicolas Suaud[1],
Laura Messio[2], Coen de Graaf[3,4] and Nathalie Guihéry[1*]

**1** Laboratoire de Chimie et Physique Quantiques, CNRS, Université de Toulouse,
118 route de Narbonne, 31062 Toulouse cedex 4, France
**2** Sorbonne Université, CNRS, Laboratoire de Physique Théorique de la Matière Condensée,
LPTMC, F-75005 Paris, France
**3** Department of Physical and Inorganic Chemistry, Universitat Rovira i Virgili,
Marcelli Domíngo 1, 43007 Tarragona, Spain
**4** ICREA, Pg. Lluís Companys 23, 08010 Barcelona, Spain

⋆ nathalie.guihery@irsamc.ups-tlse.fr

## Abstract

Herbertsmithite is known as the archetype of a $S = 1/2$ nearest-neighbor Heisenberg antiferromagnet on the kagomé lattice, theoretically presumed to be a quantum gapless spin liquid. However, more and more experiments reveal that the model suffers from deviations from the ideal one, evidenced at very low temperatures. This detailed *ab initio* study focuses on two such deviations that have never been quantitatively calculated: the anisotropic exchange interactions and the Heisenberg exchange with extra-plane magnetic impurities. The Dzyaloshinskii-Moriya interaction is found to have an in-plane component almost three times larger than the out-of-plane component, but typically obviated in theoretical studies. Moreover, it is shown that the extra-plane magnetic impurities have a strong ferromagnetic interaction (approximatively one third of the main exchange $J_1$) with the kagomé magnetic sites. Combined with an estimated occurrence of these magnetic impurities of ∼ 15%, the present results indicate that two-dimensional magnetic models only describe part of the physics.

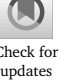

# 1 Introduction

The field of frustrated magnetism, in its ongoing search for new layered Mott insulators, remains fascinated by a now quite old one: Herbertsmithite $ZnCu_3(OH)_6Cl_2$ [1, 2]. In a first approximation, it is described by one of the most puzzling, yet extremely simple Hamiltonians: the Heisenberg model, which consists in magnetic exchange couplings of $S = 1/2$ spins on neighboring sites of a kagomé lattice.

Already at the classical level, this model remains unordered down to zero temperature (classical spin liquid), with an extensive ground state degeneracy due to a flat band of excitations [3]. This behavior is expected to persist in the quantum case. Thermal and quantum fluctuations have been extensively studied, both at the classical [4–8] and at the quantum level [9, 10], as they are expected to favor certain states, such as the coplanar ones, caused by what is known as the order by disorder effect.

At the quantum level, the exact nature of the $S = 1/2$ kagomé antiferromagnet is today still an open question, which has stimulated many theoretical and numerical studies (high temperature series expansions [11, 12], exact diagonalizations [11, 13], tensor network methods [14–16], mean-field approaches [17], variational studies [18, 19]...). After a time when the balance tipped towards a gapped spin liquid, the current trend is towards an algebraic spin liquid ground state, with no gap. The U(1) Dirac spin liquid is one such serious candidate.

Back to Herbertsmithite, the excitement of having a compound that realizes such an interesting theoretical model has led to increasingly advanced synthesis methods (high quality crystals [20]) and measurements (specific heat under high magnetic fields [21], NMR magnetic susceptibility [22, 23], thermal conductivity, magnetic structure factors via neutron scattering [24]). The precision achieved today highlights the inevitable deviations of the compound from the ideal theoretical model.

Many different deviations can occur, and their effect is inevitably strong due to the high density of low energy states. For example, further neighbor interactions lift the classical degeneracy: second neighbors favor the $\mathbf{q} = (0, 0)$ or $\sqrt{3} \times \sqrt{3}$ long-range order, while interactions beyond second neighbors open a wide range of exotic classical orders, including chiral ones [25, 26], some of which were eventually realized in other kagomé compounds [27, 28].

The most widely discussed deviation is the Dzyaloshinskii-Moriya (DM) interaction, experimentally detected by ESR [29–31] and encoded in the vector $\mathbf{d}_{ij}$. Its origin is relativistic as well as that of the symmetric tensor of anisotropy of exchange $\bar{\bar{D}}_{ij}$. Once included, the

anisotropic spin Hamiltonian reads:

$$H = \sum_{\langle i,j \rangle} \left( J_1 \mathbf{S}_i \cdot \mathbf{S}_j + \mathbf{S}_i \cdot \bar{\bar{D}}_{ij} \cdot \mathbf{S}_j + \mathbf{d}_{ij} \cdot \mathbf{S}_i \wedge \mathbf{S}_j \right), \tag{1}$$

where $i$ and $j$ are nearest-neighbor magnetic sites. When strong enough, an out-of-plane DM vector $\mathbf{d}_{ij}^\perp$ induces a $\mathbf{q} = (0,0)$ in-plane magnetic order, as in the cases of the $YCu_3(OH)_6Cl_3$ and $Cs_2Cu_3SnF_{12}$ compounds [32, 33], while an in-plane DM vector $\mathbf{d}_{ij}^\parallel$ induces weak ferromagnetism [34]. Theoretical investigations considering the DM interaction indicate that magnetic order appears for $|\mathbf{d}_{ij}^\perp/J_1| \gtrsim 0.1$ for out-of-plane vectors [17, 35–38], suggesting a smaller value for Herbertsmithite where no long-range magnetic order is observed when the temperature tends to zero. The in-plane DM component is expected to be even smaller and is mostly neglected in theoretical studies, with notable exceptions [39, 40]. Note that this component significantly complicates the problem as the total spin in the out-of-plane direction is no longer conserved.

Another relevant perturbation is the occupation disorder [41], which has important consequences. Two types of substitutions occur in Herbertsmithite: the first are magnetic vacancies within the kagomé plane, where $Cu^{2+}$ ions are replaced by $Zn^{2+}$, and the second is the opposite substitution, where magnetic impurities consisting of $Cu^{2+}$ replace inter-plane $Zn^{2+}$. While the first type seems to be quite rare, the second one has an occurrence rate of about 0.15 [23, 42, 43] with strong effects [44]. However, most theoretical studies are limited to the simpler problem of in-plane magnetic vacancies [39, 40, 45–51]. Compared to in-plane magnetic vacancies, inter-plane magnetic sites introduce two additional difficulties into theoretical studies: the dimensionality of the problem increases, and an additional, a priori unknown exchange between the impurities and kagomé magnetic sites arises.

To allow more predictive and well-oriented theoretical studies, it is important to have good estimates of the parameters of the model describing Herbertsmithite. It is known that the overall scale is $J_1 \simeq 180K$. DFT calculations have been performed [52] to evaluate eight different exchange couplings, including inter-plane ones, but DM, which has been calculated for other highly correlated materials [53, 54] was never evaluated *ab initio* up to now in Herbertsmithite. However, *ab initio* calculations combined with the effective Hamiltonian theory allow to extract all interactions of the isotropic and anisotropic spin Hamiltonian [33, 55–63].

In this article, we tackle the first *ab initio* evaluation of the symmetric anisotropy tensor and of the DM vector for Herbertsmithite, as well as the evaluation of the exchange between kagomé $Cu^{2+}$ sites and $Cu^{2+}$ inter-plane impurities.

## 2 Theory

The approach used here consists in extracting the local effective interactions of the model Hamiltonian of Eq. (1) extended to magnetic exchange interactions between non-nearest-neighbors from calculations performed on embedded clusters. The embedded cluster procedure enables correlated calculations, including relativistic treatment if required, to be carried out on small fragments immersed in a realistic representation of the environment. Two types of calculations were carried out: i) density functional theory (DFT) calculations on large clusters of different sizes and shapes to determine the main magnetic couplings between copper ions and to check the transferability of the interactions from one cluster to another, ii) *ab initio* calculations based on wave function theory (WFT) including electron correlation effects and spin-orbit couplings to determine anisotropic interactions. These last calculations, which are very costly from a computational point of view, can only be performed on small clusters

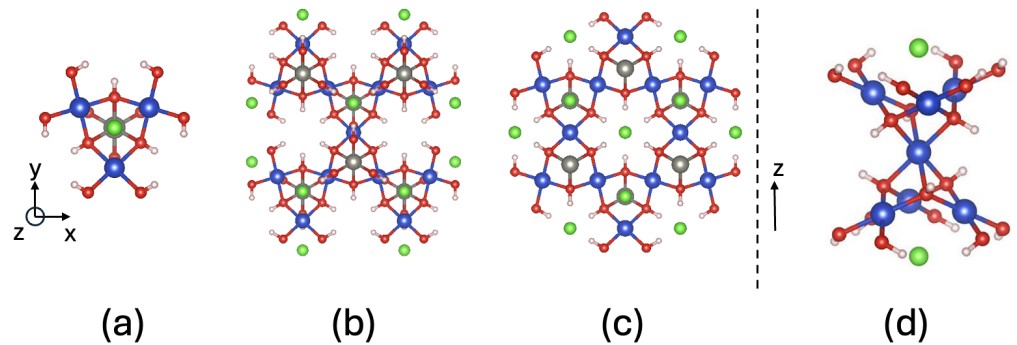

Figure 1: Clusters considered in the DFT calculations: (a) 3-copper cluster, (b) 13-copper cluster, (c) 12-copper cluster. (a), (b), (c) consider the in-plane copper only. (d) 7-copper cluster with an inter-plane $Cu^{2+}$ at the position of the $Zn^{2+}$ ion.

involving either two or three $Cu^{2+}$ ions. The Cartesian $d$-orbitals discussed below are defined according to the axes frame given in Fig. 1

## 2.1 Embedded cluster approach

The embedded cluster procedure takes into account the effects of the crystal environment by immersing a fragment of the material in a set of optimized point charges which accurately reproduces the Madelung field of the crystal. Total Ion Potentials (TIPs) were used to represent the immediate neighboring ions of atoms located on the edge of the explicitly treated cluster. The reliability of this procedure to estimate magnetic exchange parameters has been established in numerous previous studies, and has even led in some cases to the questioning of commonly used models, followed by the proposal of more appropriate ones [64–66]. The value of the main magnetic coupling $J_1$ (see Fig. 1) obtained in this local approach was compared with that obtained in a periodic calculation (see Sec. 2.4) to check the quality of the embedding. Moreover, the stability of the results can be assessed by comparing the parameters values obtained for different clusters. These verifications ensure that appropriately embedded cluster containing a small number of magnetic centers and their immediate neighbors provides a sufficiently reliable representation of the material for a correct description of the local electronic structure.

## 2.2 Broken-symmetry DFT calculations for the determination of isotropic magnetic couplings

Calculations were performed on four clusters of different sizes and shapes which are depicted in Fig. 1. The scheme of the interactions is provided in Fig. 2. Broken-spin symmetry DFT (BS-DFT) solutions have been computed by imposing the $+1/2$ or $-1/2$ value of the component $M_s$ of the spin moment in direction $z$ on each copper ion. Their energy differences were assimilated to those of the Ising Hamiltonian. Such a procedure generates different sets of equations from which the magnetic couplings can be extracted. For instance, for the 13-copper, 11 BS-DFT solutions have been computed, from which 96 sets of independent equations have been generated. Among them, 35 sets calculated from the most antiferromagnetic solutions (reversing 3 or 4 spins from the fully ferromagnetic solution) have been retained as they provide consistent values for all the interactions.

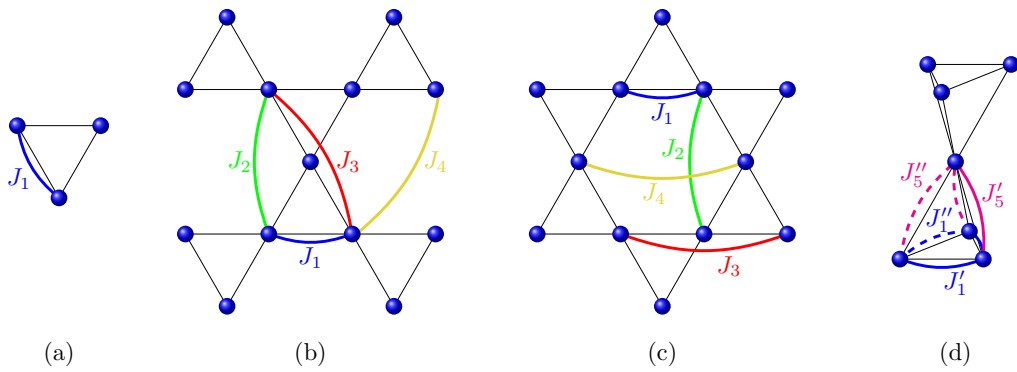

Figure 2: Scheme of the magnetic couplings for clusters (a), (b), (c) and (d) of Fig. 1. The $J_1''$ and $J_5''$ refer to pairs of $Cu^{2+}$ ions bridged by the oxygens that moved away from the impurity while for $J_1'$ and $J_5'$, the bridging oxygens have moved closer to the impurity.

One of our objectives is to determine the coupling between an in-plane magnetic center and an inter-plane impurity $Cu^{2+}$ ion located at the position of the $Zn^{2+}$. As the local symmetry of this $Zn^{2+}$ ion is $C_3$, the $Cu^{2+}$ that replaces it is Jahn-Teller active and one can expect a distortion of the oxygens (and of their bound hydrogens) coordination sphere. Such a distortion has already been studied but starting from an octahedron structure around the $Cu^{2+}$ impurity [67]. The X-Ray geometry of $C_3$ symmetry is however quite far from the octahedron (two series of angles are found $\widehat{OCuO} = 76.3°$ for oxygens bridging the impurity with two $Cu^{2+}$ of the same plane and $\widehat{OCuO} = 103.7°$ for the complementary angle). Taking the $z$ axis along the $C_3$, the Cartesian $d$-orbitals symmetries are respectively $A$ for $d_{z^2}$ and $E$ for $d_{x^2-y^2}$, $d_{xy}$, $d_{yz}$, and $d_{xz}$. The two highest in energy orbitals which are sharing the hole should have large coefficients on $d_{yz}$, and $d_{xz}$ and small coefficients on $d_{x^2-y^2}$ and $d_{xy}$, as expected from both ligand field theory and symmetry point group theory. Indeed, to be degenerate they must be of E symmetry and, again for symmetry reasons, the four orbitals can mix. Geometrical distortions induced by the Jahn-Teller effect will be studied and the magnetic couplings on the optimized structure will then be calculated. One should note that distortions will give rise to two different magnetic couplings between the $Cu^{2+}$ of the $xy$ plane, which will be denoted $J_1'$ and $J_1''$, and two different magnetic couplings between the $Cu^{2+}$ ions in the $xy$ plane and the impurity. We will denote these $J_5'$ and $J_5''$. The various couplings are depicted in Fig. 2.

## 2.3 Wave-function theory calculations for the determination of anisotropic interactions

To extract the anisotropic interactions, we have performed relativistic correlated wave-function based *ab initio* calculations. To check the computational procedure, we first reproduced the value of $J_1$, being the leading interaction in this material. For this purpose, we considered the embedded cluster (a). We performed Complete Active Space Self Consistent Field (CASSCF) calculations to account for non-dynamic correlation effects, *i.e.* the wave-functions contain at least all possible distributions of the magnetic electrons in the magnetic orbitals. The active space CAS(9,6) hold 9 electrons in 6 orbitals, namely the three magnetic orbitals of the $Cu^{2+}$ ions and 3 doubly occupied orbitals located on the bridging oxygens. These last orbitals have been optimized according to a projection procedure explained in the computational information section 2.4 [68]. Dynamic correlation effects, *i.e.* the correlation of all other electrons, are included through variational calculations using Difference Dedicated Configuration Interaction on the enlarged CAS (CAS(9,6)+DDCI1) [69], which is one of the most accurate available *ab*

*initio* methods for the calculation of exchange couplings (see Sec. 2.4).

To determine the two-body (two $Cu^{2+}$ magnetic centers) anisotropic interactions $\bar{\bar{D}}_{ij}$ and $\mathbf{d}_{ij}$, relativistic calculations have been performed on cluster (a). To check the transferability of the extracted parameters, we have considered both the cluster (a) with three $Cu^{2+}$ ions, and a dimer of $Cu^{2+}$ ions: the cluster (a) where one $Cu^{2+}$ ion has been replaced by a $Zn^{2+}$. The here-used Spin-Orbit State-Interaction (SO-SI) method [70] has been successfully used to determine anisotropic interactions in a wide variety of systems [56, 71–74]. It diagonalizes the spin-orbit matrix in the basis of all $M_s$ components of the spin states $S$ calculated at the CASSCF level. All Cu-3$d$ orbitals were introduced in the active space (see Sec. 2.4). This enlarged active space is necessary to account for the SO couplings of the ground spin-orbit free state with all excited states of the configuration. To introduce dynamic correlation on all states, we performed multireference second-order of perturbation CASPT2 [75] calculations and used the dynamically correlated energies of the spin-orbit free states as diagonal elements of the SO matrix.

To extract the symmetric anisotropic exchange tensor components and the DM vector from the *ab initio* results, we used the effective Hamiltonian theory [76,77]. This theory has been specially adapted to the extraction of anisotropic interactions and has been shown to provide very reliable values of the parameters of anisotropy in mono- and bi-nuclear complexes [56, 63, 73, 78–80]. It consists in defining a bijective relation between the target space, made up of the states calculated *ab initio*, and the model space that spans the model Hamiltonian. The effective Hamiltonian obeys the following relation:

$$H^{\text{Eff}} |\tilde{\psi}_i\rangle = E_i |\tilde{\psi}_i\rangle , \tag{2}$$

where $|\tilde{\psi}_i\rangle$ are the orthogonalized projections of the *ab initio* SO states $|\psi_i\rangle$ onto the model space and $E_i$ are the energies of the SO states calculated *ab initio*. The numerical matrix of the effective Hamiltonian can be calculated from its spectral decomposition:

$$H^{\text{Eff}} = \sum_i E_i |\tilde{\psi}_i\rangle \langle\tilde{\psi}_i| , \tag{3}$$

and then compared to the analytical matrix of the model Hamiltonian. The extraction is based on a term-by-term comparison of numerical matrix elements of the effective Hamiltonian and analytical elements of the model Hamiltonian of Eq. (1) that reads:

$$H = \begin{pmatrix} \frac{J_1}{4} + \frac{D_{zz}}{4} & \frac{D_{xz} - iD_{yz}}{2\sqrt{2}} & \frac{D_{xx} - D_{yy} - 2iD_{xy}}{4} & \frac{d_y + id_x}{2\sqrt{2}} \\ \frac{D_{xz} + iD_{yz}}{2\sqrt{2}} & \frac{J_1}{4} - \frac{D_{zz}}{4} + \frac{D_{xx} + D_{yy}}{4} & -\frac{D_{xz} - iD_{yz}}{2\sqrt{2}} & -\frac{id_z}{2} \\ \frac{D_{xx} - D_{yy} + 2iD_{xy}}{4} & -\frac{D_{xz} + iD_{yz}}{2\sqrt{2}} & \frac{J_1}{4} + \frac{D_{zz}}{4} & \frac{d_y - id_x}{2\sqrt{2}} \\ \frac{d_y - id_x}{2\sqrt{2}} & \frac{id_z}{2} & \frac{d_y + id_x}{2\sqrt{2}} & -\frac{3J_1}{4} - \frac{D_{zz}}{4} - \frac{D_{xx} + D_{yy}}{4} \end{pmatrix} , \tag{4}$$

in the basis $\{|T^+\rangle, |T^0\rangle, |T^-\rangle, |S\rangle\}$, where $|T^+\rangle, |T^0\rangle, |T^-\rangle$ are the lowest spin-orbit free triplet states of respectively $M_s = 1$, 0 and -1 and $|S\rangle$ is the lowest spin-orbit free singlet state; $J_1$ is the isotropic magnetic exchange. $D_{xy}$, $D_{xz}$ and $D_{yz}$ are the components of the symmetric tensor of exchange anisotropy $\bar{\bar{D}}_{ij}$ and $d_x$, $d_y$ and $d_z$ the $x$, $y$ and $z$ components of the DM vector. The DM vector components are directly extracted from the numerical matrix elements between the singlet and the three $M_s$ components of the triplet. One may note that these components can be expressed as functions of the components of the antisymmetric tensor $\bar{\bar{T}}_{ij}$ of the operator $\mathbf{S}_i \cdot \bar{\bar{T}}_{ij} \cdot \mathbf{S}_j$ which is identical to the operator $\mathbf{d}_{ij} \cdot \mathbf{S}_i \wedge \mathbf{S}_j$ provided that $d_x = T_{yz}$, $d_y = -T_{xz}$ and $d_z = T_{xy}$. This last remark will be useful for identifying non-zero components of the vector for symmetry reasons.

The symmetric exchange tensor is determined from the numerical interactions between the triplet components. After diagonalizing the resulting matrix of the tensor and imposing a zero trace (*i.e.* incorporating the trace in the isotropic magnetic coupling), the symmetric exchange reduces to two terms only: the axial and rhombic parameters $D$ and $E$ which are given by:

$$D = D_{ZZ} - \frac{D_{XX} + D_{YY}}{2}, \tag{5}$$

$$E = \frac{D_{XX} - D_{YY}}{2}, \tag{6}$$

where $X$, $Y$ and $Z$ are the proper magnetic axes of the symmetric tensor, in which the tensor is diagonal and which obey the convention rules: $Z$ is such that $D_{ZZ}$ is the most different diagonal element. It is a common practice to choose $X$ such that $E$ is positive and we will follow this convention here.

For the extraction on a triangular fragment involving three $Cu^{2+}$ ions, the numerical effective Hamiltonian matrix was built in the basis of eight spin-orbit-free functions: the four $M_s = -3/2, -1/2, 1/2, 3/2$ components of the quadruplet and the two $M_s = -1/2, +1/2$ components of the two doublet states. It allows to determine the three symmetric and antisymmetric (DM) tensors of the three couples of $Cu^{2+}$ ions.

## 2.4 Computational information

The geometrical structure for all heteroatoms has been taken from the X-Ray study published in reference [81].

Concerning the embedding, effective Core Potential (ECP) (for DFT calculations) and *ab initio* model potentials (AIMP) (for WFT) have been introduced to represent the ions close to the atoms of the cluster. We checked the accuracy of the embedding by comparing the B3LYP results for the $J_1$ interaction in the embedded cluster and in periodic calculations performed with the Crystal code [82] also using the B3LYP [83,84] functional. The $J_1$ values are in perfect agreement: 240 K for periodic and 239 K for embedded cluster. As we will see below, these B3LYP values slightly overestimate the coupling but demonstrate the adequacy of the material model adopted in this study. Further details on the embedded cluster method can be found in reference [85].

For the DFT cluster calculations, we used the ORCA code [86] with the def2-TZVP (Triple $\zeta$ + polarization) basis set [87] for all atoms. The geometry optimizations around the interplane $Cu^{2+}$ impurity were performed using the B3LYP functional which is known to provide accurate structures in transition metal complexes and oxides.

For the magnetic couplings calculations, the $\omega$B97X-D3 functional [88] has been preferred to the B3LYP one as it has been shown to provide better values [89].

The WFT calculations were carried out with the MOLCAS [90] and CASDI [91] codes. We used extended basis sets of ANO-RCC type [92, 93] ($6s5p3d2f$ for Cu and Zn, $4s3p2d$ for O, $4s3p1d$ for Cl and $2s1p$ for H). These basis sets are of similar accuracy to the def2-TZVP ones but their use was necessary for the relativistic calculations in the MOLCAS code. DDCI3 calculations on the minimal active space (CAS(3,3) for the trimer) are usually recommended to calculate magnetic couplings. Unfortunately, in this material, the DDCI3 value of $J_1$ is underestimated ($\sim 80K$) due to the very important role of the bridging oxygens in the antiferromagnetic contribution to the coupling. We have therefore adopted an alternative method which consists in optimizing the bridging oxygen $p$ orbitals as explained in detail in reference [68]. From the set of orbitals that was determined in the CAS(3,3)SCF calculations for the trimer, we projected the three vectors in the space of the inactive orbitals such that three orbitals are generated that are concentrated on the bridging ligands and whose shape is

**SciPost**         SciPost Phys. Core 8, 092 (2025)



Figure 3: CAS(4,3)SCF active orbital calculated on a fragment involving two in-plane copper ions and the optimized oxygen orbital (left) and the two magnetic orbitals (right).

optimal for interaction with the active orbitals. The rest of the inactive orbitals was orthogonalized to these three projected orbitals. The so-obtained $p$ ligand orbitals were then added to the active space to define a CAS(9,6). DDCI1 calculations were then performed on the top of the active space CAS(9,6) enlarged with the bridging $p$ orbitals of the oxygen. One may note that the DDCI1 version of the CASDI code also contains some double excitations with respect to the CAS which consist in a single excitation of different spin and/or symmetry inside the active space combined with a single excitation also changing the spin and/or the symmetry in the inactive spaces. These excitations which are not accounted for in standard codes are responsible for the charge and spin polarization effects which are crucial for the calculation of magnetic couplings [55].

To calculate the anisotropic interactions, enlarged active spaces containing all $d$ orbitals and their electrons were considered, namely CAS(18,10) for the bi-nuclear fragment of $Cu^{2+}$ and CAS(27,15) for the tri-nuclear one. The orbitals were optimized in a state averaged procedure for a well-balanced treatment of all excited states. Dynamic correlation was introduced at the second–order of perturbation using the CASPT2 method. The DFT orbitals are known to be too delocalized. In order to have a clear representation of the magnetic orbitals, a CAS(4,3)SCF calculation was performed on a bi-nuclear fragment to get a picture of the magnetic orbitals well-localized on just two copper ions. Their orientation and decomposition on metals and ligands are identical to the naked eye, to those obtained with the CAS(9,6)SCF obtained for the tri-nuclear fragment. Similarly with the aim of a clear visualization, CAS(4,4)SCF calculations were performed on a tetra-nuclear fragment involving three in-plane and one inter-plane copper ions.

## 3 Results and discussion

### 3.1 Isotropic couplings

In order to check the valididy of the DFT calculations that will be performed on large clusters, we have first computed the main isotropic exchange interaction $J_1$ of the model using wave function based calculations on cluster (a). The interaction of $J_1 = 181.3$ K (see Tab. 1) is in perfect agreement with values published in the literature and with the one obtained with the DFT method for the same cluster, confirming the reliability of the $\omega$B97X-D3 functional. The antiferromagnetic nature of this magnetic exchange is due to the super-exchange mechanism occurring through the bridging oxygen. Fig. 3 illustrates the contribution of the oxygen $2p$ orbital to the essentially $3d$ magnetic orbitals of the copper ions.

For fragment (d), the geometry optimization converges on a distorted structure where the two oxygen atoms (and their attached hydrogens) in the $yz$ plane have moved apart (the distances are $d_{Cu-O} = 2.23$Å instead of $d_{Zn-O} = 2.11$Å, while the other four oxygens have moved closer to $Cu^{2+}$ impurity ($d'_{Cu-O} = 2.04$Å). The lift of degeneracy of the ground state results from that of the $d$ orbitals, the essentially $d_{yz}$ one being stabilized by the distancing of

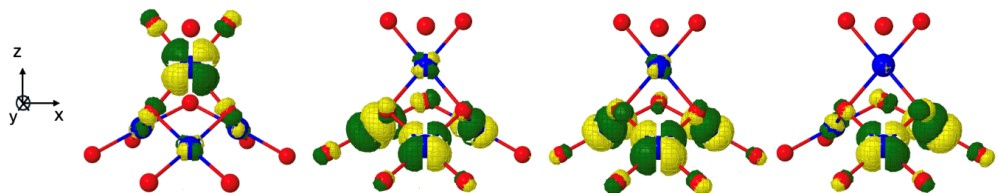

Figure 4: CAS(4,4)SCF active orbitals calculated for a tetra-nuclear fragment constituted of three in-plane (below) and an inter-plane (above) copper ions.

the ligands in the $y$ direction, in accordance with the ligand field theory. The magnetic orbital is hence of $d_{xz}$ nature with a lower contribution of the $d_{xy}$ orbital. The stabilization energy in the distorted geometry in comparison to that of $C_3$ symmetry is of 2626 K.

Table 1 reports all the computed isotropic exchange couplings obtained for the four clusters of Fig. 1. One should note the good transferability of the interaction values when increasing the size or changing the shape of the clusters (comparison of clusters (a), (b) and (c)). The antiferromagnetic $J_2$ and ferromagnetic $J_3$ and $J_4$ interactions are very weak, in line with the long distances between the $Cu^{2+}$ ions involved in these interactions. Note that the $J_4$ coupling is particularly weak despite the distance between the copper ions being identical to that of $J_3$. This is because the interactions between the copper ions involved in the exchange pass through one copper (and its surrounding oxygens) for $J_2$ and $J_3$, whereas they pass through two copper ions in the case of $J_4$.

The most important result of this series of cluster calculations is the high ferromagnetic value of $J_5'$ and $J_5''$. One may note that there are four $J_5''$ and two $J_5'$ between the inter-plane impurity and the in-plane $Cu^{2+}$ ions with which it interacts, which would result in a mean value $\bar{J}_5 = \frac{4J_5'' + 2J_5'}{6} = -64.5$ K. Concerning the main interaction $J_1$, it is the opposite: there are two $J_1'$ and one $J_1''$ which gives a mean value of $\bar{J}_1 = 149.6$ K. Around the impurities, these two types of interactions are of the same order of magnitude which should impact the collective properties of the whole material. To obtain insight into the ferromagnetic character of the inter-plane couplings, we have plotted in Fig. 4 the magnetic orbital optimized using a CAS(4,4)SCF calculation on a tetra-nuclear embedded fragment (obtained by removing the three uppermost copper ions of cluster (d) to get a clearer view). The nature of the coupling is roughly determined by three ingredients: (i) the orbital overlap increasing the kinetic exchange and hence the antiferromagnetic character of the interaction; (ii) the direct exchange, a purely ferromagnetic interaction; and (iii) the spin polarization, that may favor ferro- or antiferromagnetism depending on the system [55]. The three active orbitals on the right of Fig. 4 are strongly delocalized on the three in-plane copper ions. On the contrary, the orbital (left) localized on the impurity shows a very small overlap with the orbitals of the in-plane copper ions, excluding a large kinetic exchange contribution. The direct exchange integral is generally quite small between distant atoms, and therefore, large values of $J_5'$ and $J_5''$ must be attributed to spin polarization. The appearance of such large interactions calls into question the validity of considering this material as two-dimensional.

## 3.2 Anisotropic interactions

All anisotropic parameters are reported in Table 2. One may first note the good transferability of the parameters extracted from either bi-nuclear or tri-nuclear calculations. The anisotropic exchange values of $D$ and $E$ are very small and will have only very little impact on the magnetic properties of the system. The proper axes $X$, $Y$ and $Z$ of this tensor are given in Fig. 5.

The DM vector components, on the contrary, are non-negligible. The non-zero components

Table 1: $\omega$B97X-D3/def2-TZVP values of the isotropic exchange interactions and Cu-Cu distances for the 4 clusters of Fig. 1.

| $J_i$ (K) | (a)WFT | (a) | (b) | (c) | (d) |
|---|---|---|---|---|---|
| $J_1$ | 181.3 | 178.0 | 191.2 | 181.0 | – |
| $J_1'$ | – | – | – | – | 92.9 |
| $J_1''$ | – | – | – | – | 262.9 |
| $J_2$ | – | – | 0.5 | 0.4 | – |
| $J_3$ | – | – | -1.1 | -1.0 | – |
| $J_4$ | – | – | -0.1 | -0.2 | – |
| $J_5'$ | – | – | – | – | -47.5 |
| $J_5''$ | – | – | – | – | -73.0 |

Table 2: In-plane $|\mathbf{d}_{ij}^{\parallel}|$ and out-of-plane $|\mathbf{d}_{ij}^{\perp}|$ components of the DM vector, total DM magnitude $|\mathbf{d}|$, axial $D$, and rhombic $E$ anisotropic exchange parameters (in Kelvin) extracted from bi-nuclear and tri-nuclear fragment calculations.

| Fragment | $|\mathbf{d}_{ij}^{\parallel}|$ | $|\mathbf{d}_{ij}^{\perp}|$ | $|\mathbf{d}|$ | D | E |
|---|---|---|---|---|---|
| Bi-nuclear | 4.73 | 1.70 | 5.03 | -0.46 | 0.13 |
| Tri-nuclear | 4.73 | 1.78 | 5.05 | -0.51 | 0.16 |

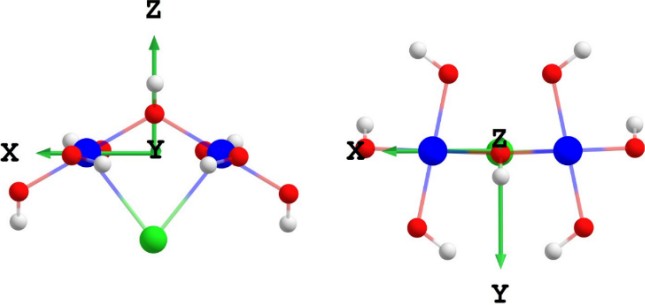

Figure 5: Two views of the proper magnetic axes of the symmetric tensor of anisotropy calculated for a bi-nuclear fragment. The $Z$ axis points toward the bridging oxygen while $X$ is the inter-nuclear axis.

of the vector can be anticipated from symmetry point group theory applied to the dimer under consideration. The dimer represented in Fig. 3 has a $C_s$ symmetry point group, $(xy)$ being the symmetry plane. In references [80,94], symmetry rules predict that the non-zero components of the antisymmetric tensor are $T_{yz}$ and $T_{xz}$ in the axis frame where the plane of the $C_s$ group is $xy$. As the reflection plane is $(yz)$ in our axes frame, the non-zero values should be $T_{xz} = -d_y$ and $T_{xy} = d_z$ ($x$ and $z$ have been exchanged). In the bi-nuclear calculations, we found $d_x = 0$, $d_y = 4.73$K and $d_z = 1.70$K. Our results are hence in agreement with the symmetry rules predicted for the $C_s$ symmetry point group. One should note that, in the tri-nuclear calculation, the DM vectors of the two pairs of $Cu^{2+}$ ions for which $(yz)$ is not the local symmetry plane exhibit three non-zero components. The $x, y, z$ components (in Kelvin) of these vectors are $d_{23} = (4.10, 2.36, -1.78)$, $d_{31} = (-4.10, 2.36, -1.78)$. For this reason and in order to compare the results obtained for a bi-nuclear fragment and the tri-nuclear one, we have chosen to define an in-plane $xy$ component $|\mathbf{d}_{ij}^{\parallel}|$ and an out-of-plane one $|\mathbf{d}_{ij}^{\perp}|$ (along $z$). Note that the in-plane component $|\mathbf{d}_{ij}^{\parallel}|$ of the DM vector is larger than the out-of-plane one $|\mathbf{d}_{ij}^{\perp}|$. This *a priori* surprising result can, however, be rationalized. Indeed, as demonstrated analytically

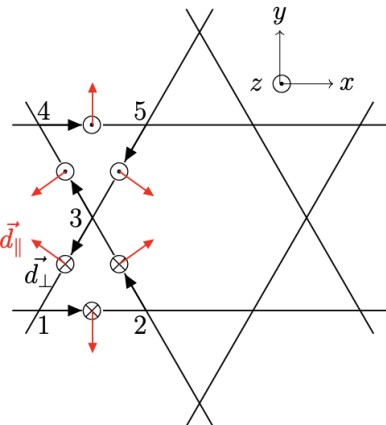

Figure 6: Picture of the DM vectors on two neighbor triangles of the lattice. The black arrows between the magnetic centers indicate the order of the first and second spins in the vector products of the DM interactions. The out-of-plane components alternate from one triangle to its neighbors and point in the opposite direction to the position of the oxygens. The angle between the in-plane components of the vectors within the triangle is 120°.

in [57], the physical origin of the DM vector is the hybridization of the metal's Cartesian $d$-orbitals (here chosen for the bi-nuclear fragment of Fig. 3). The mixing of the $d$ orbitals conditions the nature of this interaction. The analytical derivation presented in reference [57] demonstrates that

- a mixing between $d_{x^2-y^2}$ and $d_{xy}$ or between $d_{xz}$ and $d_{yz}$ generates a $d_z$ component,

- a mixing between $d_{x^2-y^2}$ and $d_{xz}$ or between $d_{xy}$ and $d_{yz}$ generates a $d_y$ component,

- a mixing between $d_{x^2-y^2}$ and $d_{yz}$ or between $d_{xy}$ and $d_{xz}$ generates a $d_x$ component.

The contribution of the out-of-plane $d_{xz}$ and $d_{yz}$ Cartesian orbitals in the magnetic orbitals is clearly evidenced in Fig. 3 and can be appreciated by their coefficients in the magnetic orbitals. The expression of one of these magnetic orbitals on the Cartesian $d$-orbitals of one copper ion is: $0.4889 d_{x^2-y^2} + 0.2120 d_{xz} - 0.2841 d_{xy} - 0.3666 d_{yz} +$ coefficients on the ligands orbitals. This orbital mixing rationalizes the strong in-plane component of DM vector. Its orientation is depicted in Fig. 6, where we may note that its out-of-plane component $|\mathbf{d}_{ij}^{\perp}|$ oscillates between below and above the $(xy)$ plane of copper ions from one triangle to the next, following the alternating positions above and below of the oxygens.

## 4 Summary and conclusions

The embedded cluster model is used to evaluate isotropic and anisotropic exchange interactions in Herbertsmithite, characterized by $S = 1/2$ spins located on a kagomé lattice. The leading magnetic interaction is the in-plane nearest neighbor isotropic exchange. This interaction is antiferromagnetic in nature. Its strength is experimentally evaluated to $J_1 \sim 180$ K in a Heisenberg Hamiltonian $H = \sum_{\langle i,j \rangle} \mathbf{S}_i \cdot \mathbf{S}_j$, $i$ and $j$ being nearest neighbors. Both the DFT and WFT values are in close agreement with this experimental result, validating the material model and the electronic structure methods used. This is in line with many previous studies of exchange interactions in similar compounds, using the same computational techniques.

Second-neighbor in-plane interactions (and beyond) are all very small and there is no need to include these interactions in model studies. Triggered by the experimental evidence for a substantial number of magnetic impurities located between the $S = 1/2$ kagomé planes, we have also calculated the isotropic exchange between a regular in-plane $Cu^{2+}$ center and a copper ion replacing an inter-plane $Zn^{2+}$. As the local symmetry around the impurity is $C_3$, the two highest in energy orbitals of the $Cu^{2+}$ ion are degenerate and the site is Jahn-Teller active. The geometry optimization shows a distortion in which four oxygens get closer to the $Cu^{2+}$ ion and two get further away, giving rise to two different couplings $J'_5$ and $J''_5$ between the impurity and the in-plane copper ions. Similarly, two couplings $J''_1$ and $J''_1$ are observed between the in-plane ions. The interactions with the impurity turns out to be ferromagnetic and far from being negligible, namely $J'_5 = -48$ K and $J''_5 = -73$ K. The commonly used models to rationalize the experimental observations only consider in-plane interactions, but this out-of-plane interaction questions the validity of a simple bi-dimensional model.

Concerning the anisotropic interactions, the results indicate that the symmetric anisotropic exchange interaction does not contribute in a significant manner to the low-energy spectrum of this material. With $D$ and $E$ values smaller than 1 K for the axial and rhombic anisotropic exchange parameters respectively, it is not expected that this interaction plays a role in the magnetic properties. This is not the case for the Dzyaloshinskii-Moriya vector (or antisymmetric tensor of anisotropy). The values extracted from the combination of *ab initio* WFT calculations and effective Hamiltonian theory are on the order of several Kelvin with the in-plane component significantly larger than the out-of-plane component. The occurrence of these two non-zero components is consistent with symmetry rules. The analysis of the magnetic orbitals shows that there is a sizeable mixing of the $d_{xy}$ and $d_{x^2-y^2}$ in plane Cartesian orbitals with the $d_{xz}$ and $d_{yz}$ ones in the magnetic orbitals, caused by the out-of-plane position of the bridging oxygen ions. This mixing was shown in previous studies to originate in plane DM interactions [57]. The three DM pseudo-vectors of the triangles formed by the $Cu_3$-O units all point in the direction opposite to the oxygen positions and result in a small net out-of-plane interaction that alternately points up and down for neighboring triangles.

Our calculations thus provide a faithful and quantitative description of the spin Hamiltonian of Herbertsmithite, that is far more complete than the widespread simple nearest neighbor kagomé antiferromagnet. We evidence the importance of two supplementary contributions: the out-of-plane DM interaction and a strong ferromagnetic exchange between the kagomé sites and the ~15% of inter-plane magnetic impurities. These two contributions to the Hamiltonian are theoretically challenging. Their effect on the physics is unclear: DM tends to favor long-range order, but only above a threshold value, and the out-of-plane component has been far less studied than its in-plane counterpart. On the other side, site disorder probably has unsuspected consequences through a cross-over between 2D and 3D physics. We hope this study will stimulate theoretical studies of this complex model.

The results derived here for Herbertsmithite are valid for a substitution rate of the Zn by Cu up to 0.66, through the Zn-paratacamite family. But for higher substitution, a structural transition has been reported [95], and the fully substituted compounds, so-called clinoatacamite, is expected to have different exchanges. A further work will be devoted to the study of this material. Another extension of this work could evaluate how the DM interactions in the triangles of the kagomé lattice are affected by the presence of a $S = 1/2$ ion completing the coordination of the bridging oxygen.

**Data availability statement:**    Data are available on request from the authors.

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
