# Peer review of "Is Herbertsmithite far from an ideal antiferromagnet? Ab-initio answer including in-plane Dzyaloshinskii-Moriya interactions and coupling with extra-plane impurities"

_SciPost Physics, doi:SciPost Phys. Core 8, 092 (2025)_

## Round 1 · Author Response

The authors thank the referees for having accepted to review this work.

Reviewer 1:

1 It is not clear from the paper why authors decided to treat isotropic and anisotropic exchange separately with different methods. Within the effective Hamiltonian approach, they employ to obtain DM interactions, they also obtain J1, which should probably be considered as their best estimates. But it is not reported in Table 2. It should also be possible to estimate J5 and the corresponding possible anisotropic interactions with out-of-plane impurity using WF-based methods to get consistent picture.

As noticed by the reviewer the best estimates are provided by WF-based methods. We agree with the reviewer and have changed the presentation of the results. We have now started the section results with the WF-based calculations. The WF results are used to validate those of DFT obtained with the ωB97X-D3 functional. A comment has been added to specify that WFT calculations provide our best estimates. Unfortunately, these methods are so computationally expensive that only very small fragments (with two or three Copper ions) can be treated. For instance, the zeroth-order CASSCF calculation to be performed on cluster (b), with the same active space than the one used for the trimer (cluster a), i.e. one orbital per Cu 2+ plus the p orbital of the bridging oxygens, would require the diagonalization of a matrix of dimension 1.9.10 12×1.9.1012. Furthermore, the CASSCF calculation typically provides 10% of antiferromagnetic coupling values and this is why we always perform Configuration Interactions like DDCI1 (with an enlarged active space) or DDCI3 (with a minimal active space) or CASPT2 on top of the reference (CAS) space. This second step which accounts for dynamical correlation is far costlier than the CASSCF step. Furthermore, it is important to perform calculations on clusters of different shapes and sizes in order to verify the transferability of the extracted interactions. Regarding the calculation of couplings with the impurity, as it is connected to six other Cu2+ (three above and three below), the minimal fragment (to obtain a well-balanced description of magnetic couplings and an impurity-centred embedding) contains 7 Cu 2+. Here again, the dynamically correlated calculation on the enlarged CAS is impossible. Please note that in the revised version, structure optimizations around the impurity have been performed and that we have now two different J5 and two different J1. Finally, as the reader could not be aware of the cost of WFT calculations, we have added the following paragraph in the introduction of the section theory to justify our choices: “Two types of calculations were carried out: i) density functional theory (DFT) calculations on large clusters of different sizes and shapes to determine the main magnetic couplings between copper ions and to check the transferability of the interactions from one cluster to another, ii) ab initio calculations based on wave function theory (WFT) including electron correlation effects and spin-orbit couplings to determine anisotropic interactions. These last calculations which are very costly from a computational point of view can only be performed on small clusters involving either two or three Cu2+ ions.”

2 In the DFT calculations of further-neighbor isotropic couplings using bigger clusters the authors opted for choosing a different basis set compare to WF-based part of the study, which is again reduces the coherency of the study.

Due to the lower cost of DFT calculations we used a larger basis than in WFT calculations. The two extended basis sets used here are of comparable accuracy. The use of ANO-RCC is compulsory with the MOLCAS code for the calculation of anisotropic interactions. Indeed, the treatment of relativistic effects in MOLCAS is performed by partitioning the relativistic effects into two parts: the scalar relativistic effects and spin-orbit coupling. This partitioning is based on the Douglas-Kroll (DK) transformation of the relativistic Hamiltonian. It is necessary to use a basis set with a corresponding relativistic contraction. MOLCAS uses the ANO-RCC basis set, which has been constructed using the DK Hamiltonian.

3 In the text authors advocate for the DDCI method to be one of the most accurate WF-based methods to treat dynamical correlations for magnetic systems. The DDCI3 indeed shows very good performance for exchange couplings. But in this study the least accurate form (DDCI1) equivalent to MRCI-S (multireference configuration interaction with single excitations) is used, which is tend to overestimate magnetic couplings. This point should not be hidden, but rather discussed in the text.

We thought (and maybe we were wrong) that entering into details was unnecessary for the physicist readers of this journal. We also performed the DDCI3 calculation on the minimal active space which is the recommended method but, as it occurs for some antiferromagnetic compounds, the value is underestimated (~80 K). As shown in the reference [65], Bordas et al, Chemical Physics 309, 259 (2005), to include an optimized p orbital of the bridging oxygen and to perform a DDCI1 calculation on this enlarged active space is an appropriate solution in such situations. It should be noted that the DDCI1 calculation performed with the CASDI code also includes some di-excited determinants, especially those that change the spin and/or symmetry inside the active space combined with an excitation out of the active space that also changes spin and/or symmetry. As shown in ref [53], Malrieu et al, Chem. Rev. 114, 429 (2014), these excitations are responsible for both the charge and spin polarization effects which are crucial for the calculation of magnetic couplings. These excitations are not accounted for in the DDCI1 of ORCA which is a true MR+Singles. We have added the following paragraph in the computational information section of our revised version: “DDCI3 calculations on the minimal active space (CAS(3,3) for the trimer) are usually recommended to calculate magnetic couplings. Unfortunately, in this material, the DDCI3 value of J1 is underestimated (~80K) due to the very important role of the bridging oxygens in the antiferromagnetic contribution to the coupling. We have therefore adopted an alternative method which consists in optimizing the bridging oxygen p orbitals as explained in detail in reference. From the set of orbitals that was determined in the CAS(3,3)SCF calculations for the trimer, we projected the three vectors in the space of the inactive orbitals such that three orbitals are generated that are concentrated on the bridging ligands and whose shape is optimal for interaction with the active orbitals. The rest of the inactive orbitals was orthogonalized to these three projected orbitals. The so-obtained p ligand orbitals were then added to the active space to define a CAS(9,6). DDCI1 calculations were then performed on the top of the active space CAS(9,6) enlarged with the bridging p orbitals of the oxygen. One may note that the DDCI1 version of the CASDI code also contains some double excitations with respect to the CAS which consist in a single excitation of different spin and/or symmetry inside the active space combined with a single excitation also changing the spin and/or the symmetry in the inactive spaces. These excitations which are not accounted for in standard codes are responsible for the charge and spin polarization effects which are crucial for the calculation of magnetic couplings.”

4 Authors mention that the orientation of the DM vector (would be the fraction of d_parallel/d_perpendicular in the paper's notation) is "a priori surprising result", but in most cases this orientation is fully determined by the local point-group symmetry of the dimer. Such analysis should be a natural part of the paper.

The referee is absolutely right: the orientation of the DM vector is predictable by symmetry. We have added a paragraph in which we show that the out-of-plane component has no reason to be zero for reasons of symmetry. Two paragraphs have been added. The first one in the Section Theory and the second one in the section “anisotropic interactions”. The first one, in section II.3 : “One may note that these components can be expressed as functions of the components of the antisymmetric tensor T́ ij of the operator S i . T́ ij . S j which is identical to the operator d ij . Si ⋀ S j provided that dx = Tyz, dy = -Txz and dz = Txy. This last remark will be useful for identifying non-zero components of the vector for symmetry reasons.” The second one in section III.2: “The non-zero components of the vector can be anticipated from symmetry point group theory applied to the dimer under consideration. The dimer represented in Figure 3 has a Cs symmetry point group, (yz) being the symmetry plane. In references [80,94], symmetry rules predict that the non-zero components of the antisymmetric tensor are Tyz and T xz in the axis frame where the plane of the Cs group is (xy). As the reflection plane is (yz) in our axes frame, the non-zero values should be T xz = - dy and Txy = dz. (x and z have been exchanged). In the bi-nuclear calculations, we found dx = 0, dy = 4.73 K and dz = 1.70 K. Our results are hence in agreement with the symmetry rules predicted for the Cs symmetry point group. One should note that, in the trimer calculation, the DM vectors of the two pairs of Cu 2+ ions for which (yz) is not the local symmetry plane exhibit three non-zero components. The x, y, z components (in Kelvin) of these vectors are d23 = (4.10, 2.36, −1.78), d31 = (−4.10, 2.36, −1.78). For this reason and in order to compare the results obtained for a bi-nuclear fragment and the tri-nuclear one, we have chosen to define an in-plane xy component |dij ∥ |and an out-of-plane one |dij⊥| (along z).

5 There are other small things: Authors say they use orbitals from small CAS(2,2)SCF and CAS(4,4)SCF calculations for illustration purposes instead of actual DFT or CASSCF calculations. Nevertheless, in the text they actually mention and analyze their shape. In this case the orbitals from the actual calculations should be employed.

The DFT orbitals are known to be too much delocalized on the ligands which make their use for analysis purpose not very easy. We now provide the CAS(4,3)SCF orbitals for the bi-nuclear fragment which are identical to those of the CAS(9,6)SCF for the tri-nuclear fragment. For visibility reasons it is more convenient to have a bi-nuclear fragment showing the orientation of the magnetic orbitals and their tails on the oxygen ligands. We now also show the optimized p orbital of the bridging oxygen. The same arguments are valid for the CAS(4,4)SCF used to describe the magnetic orbitals of the tetra-nuclear fragment.

6 In the table 2 the authors report results of the symmetric anisotropic exchange in a very rudimentary form. The D and E are not defined, moreover without the corresponding main magnetic axes the results are not complete.

We have introduced the definition of the D and E parameters in the section “Theory” and extracted the proper axes of the symmetric tensor. Figure 5 shows the axes of the tensor. The following paragraph has been introduced in section II.3. “The symmetric exchange tensor is determined from the numerical interactions between the triplet components. After diagonalizing the resulting matrix of the tensor and imposing a zero trace (i.e. introducing the trace in the isotropic magnetic coupling), the symmetric exchange reduces to two terms only: the axial and rhombic parameters D and E which are given by: D = D_ZZ −(D_XX + D_YY)/2 E = (D_XX − D_YY)/2 where X, Y and Z are the proper magnetic axes of the symmetric tensor, i.e. the axes in which the tensor is diagonal and which obey the convention rules: Z is such that D ZZ is the most different diagonal element. Quite often, X is chosen such that E is positive and this will be the case here.”

Requested changes: 1 Reason in the manuscript why anisotropic and isotropic interactions are obtained with WF-based and DFT-based calculations respectively. 2 Clearly present all couplings obtained from WF-based method.

These two points have been addressed above.

3 Possibly use consistent basis sets and fuctionals though the study.

Concerning the functionals, B3LYP is known to be very efficient for geometry optimization. The ωB97X-D3 functional is the best to treat magnetic exchanges but is unfortunately not available in periodic DFT calculations performed with Crystal.

4 Clearly state that the employed DDCI1 method is equivalent to MRCI-S and is less accurate compare to the most of the cited DDCI studies. 5 Provide symmetry analysis of the two- (and possibly three-) magnetic-site problem with symmetry-motivated direction of the DM vector

Points 4 and 5 have been addressed above.

6 Use orbitals form actual DFT and CASSCF/DDCI1 studies in the shapes Analysis.

The DFT orbitals are less accurate than those reported in the article. DDCI1 is not a method of optimization of the orbitals, the orbitals used in the DDCI1 method are the CASSCF ones.

7 Provide expressions for D and E in the Table 2 together with the associate main magnetic axes.

This is done in the revised version.

Reviewer 2

1 The sizable ferromagnetic exchange between the impurity spins and kagome planes exceeding -80 is a new interesting result that indeed questions the quasi-2D nature of herbertsmithite. However, it is difficult to reconcile this result with the Weiss temperature of -5.2 K extracted from ESR data (PRL 118, 017202 [2017]). Also averaged values obtained using DFT on locally optimized structures (Tab. 7.2 in https://nbn-resolving.org/ urn:nbn:de:bsz:14-qucosa-91976) are significantly smaller (between -10 and -20 K). In the latter case, a structural relaxation has been performed to simulate a Jahn-Teller distortion of an octahedron under Cu substitution. Fingerprints of a such distortion are also found in the experiment, e.g. in Ref. 23: "...a staggered response is certainly associated with a Jahn-Teller driven distortion leading to a displacement of the six adjacent oxygens". This makes me think that structural relaxation is a key ingredient that cannot be bypassed if a magnetic Jahn-Teller-active impurity like Cu2+ occupies a regular octahedron.

We agree with the reviewer and thank him for this crucial remark. The full study of the couplings between the in-plane magnetic ions with the impurity has been performed after geometry optimization of the oxygens (and their attached hydrogens) of the coordination sphere of the impurity Cu2+ ion. It has required many changes in the redaction of the new version. In Section II.2, we have added the following paragraph: “One of our objectives is to determine the coupling between an in-plane magnetic center and an inter-plane impurity Cu 2+ ion located at the position of the Zn 2+. As the local symmetry of this Zn2+ ion is C3, the Cu2+ that replaces it is Jahn-Teller active and one can expect a distortion of the oxygens (and their connected hydrogens) coordination sphere. Such a distortion has already been studied but starting from an octahedron structure around the Cu 2+ impurity. [64] The X-Ray geometry of C3 symmetry is however quite far from the octahedron (two series of angles are found ( OCuO ^ )=76.3 ° for oxygens bridging the impurity with two Cu2+ of the same plane and ( ^ OCuO )=103.7 ° for the complementary angle). Taking the Z axis along the C3, the Cartesian d-orbitals symmetry are respectively A for d z2 and E for d x2 − y 2, d xy, d yz , and d xz. The two highest in energy orbitals which are sharing the hole should have large coefficients on d yz , and d xz and small coefficients on d x 2 − y2 and d xy , as expected from both ligand field theory and symmetry point group theory. Indeed, to be degenerate they must be of E symmetry and, again for symmetry reasons, the four orbitals can mix. Geometrical distortions induced by the Jahn-Teller effect will be studied and the magnetic couplings on the optimized structure will then be calculated. One should note that distortions will give rise to two different magnetic couplings between the Cu2+ of the (XY) plane, which will be denoted J’1 and J”1, and two different magnetic couplings between the Cu 2+ ions in the (XY) plane and the impurity. We will denote these J’ 5 and J”5. The various couplings are depicted in Figure 2.”

In the section results III.1, we have added the following paragraph and added the new couplings to table 2. “For fragment (d), the geometry optimization converges on a distorted structure where the two oxygen atoms (and their attached hydrogens) in the (YZ) plane have moved apart (the distances are dCu-O=2.23 Å instead of dZn-O=2.11 Å, while the other four oxygens have moved closer to Cu2+ impurity (d’Cu-O= 2.04 Å). The lift of degeneracy of the ground state results from that of the d orbitals, the essentially d yz one being stabilized by the distancing of the ligands in the Y direction, in accordance with the ligand field theory. The magnetic orbital is hence of d xz nature with a lower contribution of the d xy orbital. The stabilization energy in the distorted geometry in comparison to that of C3 symmetry is of 2626 K.”

And two paragraphs later: “The most important result of this series of cluster calculations is the high ferromagnetic value of J’5 and J”5. One may note that there are four J”5 and two J’5 between the inter-plane impurity and the in-plane Cu2+ ions with which it interacts, which would result in a mean value of J5 = (4 J ' ' 5 +2 J ' 5)/6= −64.5 . Concerning the main interaction J1, it is the opposite, there are two J’1 and one J”1 which gives a mean value of J 1=149.6 K . Around the impurities, these two types of interactions are of the same order of magnitude which should impact the collective properties of the whole material. To obtain insight into the ferromagnetic character of the inter-plane couplings, we have plotted in Figure 4 one magnetic orbital derived from a CAS(4,4)SCF calculation on a tetra-nuclear embedded fragment (obtained by removing the three uppermost copper ions of cluster (d) and replacing one Cu2+ by a Zn2+ to get a clearer view).”

2 While the DM interaction reasonably turns out to be the leading anisotropy, its magnitude -- less than 3% of the Heisenberg exchange -- seems to be rather low. Note that in La2CuO4 with a very small tilt angle (3-4 degrees), the DM anisotropy amounts to about 0.6 % of the Heisenberg exchange (PRB 108, 085140 [2023]). In herbertsmithite, the neighboring CuO4 squares are significantly more tilted, giving rise to a sizable overlap between x^2-y^2 and other d-orbitals, which is also stated in the manuscript, e.g. in the caption of Fig. 4. I would naively expect a more significant enhancement of the DM component in this case. I suggest the authors benchmark their approach by calculating D1 and J1 in La2CuO4, for which experimental estimates are available and which have been extensively studied by other numerical methods.

We have performed WF based calculations of the nearest neighbor magnetic exchange and DMI in La2CuO4, using exactly the same procedure as the one followed in this article. We find J=106.6 meV and |DMI|=0.45 meV, i.e. 0.42% of the magnetic coupling. This ratio is rather close to what is found in the literature. Our ratio in Herbertsmithite which is 2.8% is more than 6 times larger which can effectively be correlated with the more tilted angle ( CuOCu ) in Herbersmithite. However, as it is shown in our article (ref 56) on the physical origin of DMI, covalency increases the value of DMI due to the difference between the singlet and triplet orbitals which is a crucial ingredient to have a sizable DMI. With a magnetic coupling of 1237 K in La2CuO4 in comparison to 181 K in Herbersmithite, it is quite clear that covalency is much more important in La 2CuO4 rationalizing the comparable values of the DMI in both compounds. Ab initio calculations based on the wave function do not involve any adjusted parameters, unlike DFT. The Hamiltonian is the exact electronic Hamiltonian in the Born-Oppenheimer approximation. MOLCAS relativistic calculations are performed by partitioning relativistic effects into two parts: scalar relativistic effects and spin-orbit coupling. This partition is based on the Douglas-Kroll (DK) transformation of the relativistic Hamiltonian. The SO-RASSI method of MOLCAS has been used in numerous studies where comparisons of anisotropic parameters with experiment values show remarkable accuracy (see, for example, Nature Chemistry, 2025, ⟨10.1038/s41557-025-01926-5⟩, or Inorganic Chemistry 2011, 10.1021/ic200506q).

Minor points:

1) This becomes obsolete if the authors perform a structural optimization, as suggested earlier, but I still wonder why the structural input is taken from Ref. 74, which is a lab XRD study on natural samples. There are more recent and more detailed structural studies on synthetic samples, e.g. JACS 127, 13462 [2005], JACS 132, 16185 [2010], Appl. Phys. Lett. 98, 092508 [2011].

We have performed a geometry optimization around the Jahn-Teller active interplane Cu 2+ ion and redone the study with the optimized structure as explained above. We have also compared the internal coordinates of the here used geometrical structure with those mentioned by the reviewer and found very similar internal coordinates, with a maximum difference of 10 –2 Å in one bond distance (Zn-O) all the others being of 10–3 Å.

2) The naming convention for orbitals is somewhat vague. My understanding is that "Cartesian d-orbitals" are standard d-orbitals in some global Cartesian frame with the x axis along a Cu-Cu bond; "d^2-y^2", "dxz" etc. presumably refer to a local coordinate frame of CuO4. Please explain how these frames are defined.

The Cartesian d-orbitals are expressed in the axes frame of figure 1. A sentence has been added to clarify. End of the first paragraph of section theory: “The Cartesian d-orbitals discussed below are defined according to the axes frame given in Figure 1.”

3) Note that there exist other numerical DFT-based methods for calculations of DM interactions, such as the Green's-function methods (PRB 71, 184434 [2005]) and the energy-mapping approach (Dalton Trans. 42, 823 [2013]). Both have been applied to several Cu2+-containing oxides.

The authors thank the reviewer for these additional references. These articles have been added to the bibliography.

Requested changes

  1. Perform structural optimization of clusters with impurity Cu atoms.

A full study has been performed with optimized geometries.

  1. Benchmark the method used to estimate DM interactions by performing a similar analysis for a well-studied material such as La2CuO4.

We have performed calculations on the La 2CuO4 materials and found values close to those found in the literature.

---

## Round 1 · List of Changes

Paragraphs have been added, given in the answer to referees.

---

## Editorial Decision

published